# Benefit finding and well-being over the course of the COVID-19 pandemic

Jessie B. Moore[1]*, Katharine C. R. Rubin[1], Catherine A. Heaney[1,2]

1 Stanford Prevention Research Center, Stanford School of Medicine, Stanford, California, United States of America, 2 Department of Psychology, Stanford University, Stanford, California, United States of America

* jbmoore@stanford.edu

## Abstract

This study focuses on understanding benefit finding, the process of deriving growth from adversity, and its relationship to well-being amidst the COVID-19 pandemic. Participants (n = 701) completed online surveys at 1, 3, 6, and 12 months after a shelter-in-place mandate was announced in California, USA. Identifying as female or of Asian descent, having a supportive social network, and reporting more distress were associated with higher levels of general benefit finding at all data collection points, while other demographics were not. Benefit finding exhibited small but statistically significant associations with two measures of well-being. Understanding the extent to which various groups of people experience benefit finding during ongoing adversity and how such benefit finding is associated with well-being may help to promote mental health during a collective trauma like the COVID-19 pandemic.

## Introduction

As the world has been experiencing the COVID-19 pandemic, the popular media has focused on both the negative and positive ways in which people's lives have changed as a result. The pandemic can be considered a world-wide collective trauma [1], defined by Hirschberger as a "cataclysmic event that shatters the basic fabric of society" [2 p. 1]. Research has documented the toll that the pandemic has had on both physical and mental health, with further knowledge accruing on a daily basis [3–5]. Amidst all of the suffering, some people have coped more resiliently than others and some have been able to derive some benefit from their pandemic experiences [6]. In this paper, we examine the extent to which benefit finding has been experienced during different phases of the pandemic, the demographic characteristics of those most likely to experience benefit finding, and the role of distress and social relationships in this experience. Lastly, we explore the relationship between benefit finding and well-being.

### Nature of benefit finding

While the concept of experiencing transformative positive change in response to trauma is far from new, the last two decades have seen a large uptick in research on this topic. The term post-traumatic growth (PTG) has been used to describe new ways of thinking, feeling and behaving that people may experience after surviving a trauma [7]. As its name clearly implies, PTG connotes changes that occur after a traumatic event rather than during the event, often

**Data Availability Statement:** Data relevant to this study are available from the Harvard Dataverse at doi:10.7910/DVN/GNIOTF.

**Funding:** Initial foundational funding for the Stanford Wellness Living laboratory (WELL) was

provided by Amway via an unrestricted gift through the Nutrilite Health Institute Wellness Fund to Stanford University. Preparation of this manuscript was supported in part by the Stanford Thailand Research Consortium.

**Competing interests:** The authors have declared that no competing interests exist.

after the person has had the opportunity to reflect on the experience. However, during traumatic events of long duration and multiple phases, it is possible that such reflection may occur during the experience of the trauma. Thus, the experience of benefit finding may begin before the traumatic event is over. In this study, we explore the experience of benefit-finding at different stages of the COVID-19 pandemic when the risk of infection and mandates for protective behavior were waxing and waning.

Various types of benefits or growth have been documented in the literature. These include new perceptions of one's abilities, strengths and resilience in the face of challenges; new ways of thinking about and relating to others; a greater appreciation for life; explorations of new life possibilities or pathways; and changes in one's spiritual views [8–10]. These changes are meant to be transformative, indicating meaningful and important transitions. The extent to which these changes endure is not well-demonstrated empirically, but the theoretical expectation is that they are long-lasting [7].

## When and by whom benefit finding is likely to be experienced

The current literature offers inconsistent findings in terms of the relationships between demographic characteristics and benefit finding. There is some evidence that women and people of color are more likely to experience benefit finding than are men and white people, but the findings for other demographic characteristics (e.g., marital status, age, and educational attainment) are mixed [8, 9, 11].

There is little research evidence suggesting that different types of traumas are likely to elicit varying quantities or qualities of benefit finding. In a collective trauma such as the COVID pandemic, people may experience both personal suffering (e.g., through personal infection, loss of a loved one, loss of one's job) as well as societal suffering (e.g., over-run medical care establishments, high unemployment rates, increased preventable mortality). A few studies have demonstrated that in response to collective traumas such as natural disasters, the likelihood of experiencing changes in how one thinks about or relates to others is greater than the likelihood of other potential perceived benefits [12]. This may be because the salience of others' coping and helping behaviors is greater when everyone in the community has to respond to the trauma.

The availability of supportive relationships has been associated with benefit finding [13]. Having supportive interactions with others may facilitate disclosure of one's personal suffering and discussion of societal suffering more generally. Such discussions may generate more of an ability to reflect on one's experience of the trauma and to identify positive sequelae. They may also help to generate new or revised narratives about the nature of the trauma and the effectiveness of coping alternatives. Lastly, receiving or even simply observing the provision of instrumental support by others may catalyze new beliefs about personal and collective strengths. Thus, **we hypothesize that individuals that report stronger support networks will be more likely to experience subsequent benefit finding**.

The levels of distress reported by people during the pandemic are likely to vary due to both personal and societal suffering. Because benefit finding is thought to be catalyzed by the distress caused by the trauma [8], **we hypothesize that people who report higher levels of distress (regardless of whether the source of the distress is personal or societal) will be more likely to experience subsequent benefit finding**.

The time trajectory of benefit finding has two component parts. First, how long after the trauma is benefit finding most likely to occur? Second, once experienced, how long will the positive changes inherent in benefit finding be maintained? Neither of these questions have clear answers in the extant literature. Much of the literature has been cross-sectional, thus not

allowing strong inferences to be made about time trajectories. The few longitudinal studies have found a variety of different trajectories. Studies of veterans [14], breast cancer survivors [15], and young adults with cancer [16] found a fair amount of stability over time in terms of benefit finding (i.e., participants with high or low levels of benefit finding maintained those levels over time). However, there were also participants for whom the experience of benefit finding waxed and waned over time. In this study, we will explore the trajectory of benefit finding during a year of the pandemic. **We hypothesize that the amount of benefit finding will increase as the pandemic progresses and individuals have more opportunity to gain insight about their experiences.**

### Benefit finding and well-being

The relationship between benefit finding and well-being has been much explored, but has yielded quite mixed results [7–9]. This may be due to variations among the research studies in terms of the nature of the traumas experienced, the timing of the measurements, and the conceptualization (and thus operationalization) of well-being. In their meta-analysis, Helgeson and her colleagues [8] found the strongest association between benefit finding and well-being when measures of positive well-being (i.e., positive affect, self-esteem, life satisfaction) were used. They found no relationship between benefit finding and subjective reports of physical health nor with comprehensive, multi-faceted measures of well-being such as quality of life. In this present study, we use both a novel broad multi-faceted measure of well-being and a more limited, well-validated measure of positive well-being. **We hypothesize a positive relationship between benefit finding and subsequent well-being, particularly for the latter measure of well-being.** For our broad multi-faceted measure of well-being, our analysis is more exploratory.

## Methods

### Overall design

Participants were recruited from an online data registry, the Stanford WELL for Life initiative [17]. T. Those who provided data at all time points were included in this study. The study was approved by the Stanford University Institutional Review Board. Informed written consent was obtained from all participants prior to the start of the study.

### Context

The majority of participants (81.0% of our sample) resided in the San Francisco Bay Area, where a regional SIP order was imposed on March 17th 2020, mandating closure of all non-essential businesses and directing individuals to shelter at their place of residence. Essential services included grocery stores, health care facilities, pharmacies, gas stations, convenience stores, banks and laundromats. Restaurants were only permitted to offer delivery and takeout services. Nonessential gatherings of any size were banned, and residents were instructed to only leave their houses for groceries and supplies, care for family members, and outdoor exercise.

Many restrictions remained in place through the end of May 2020, and some continued throughout 2021 [18]. Restrictions in California were constantly being put in place and then lifted as rates of coronavirus infection fluctuated. For example, when first announced, masks were required in high-risk indoor settings such as grocery stores, but soon were also required outdoors when distancing was not possible (less than 6 feet), and eventually mandatory for all indoor and outdoor activities [19].

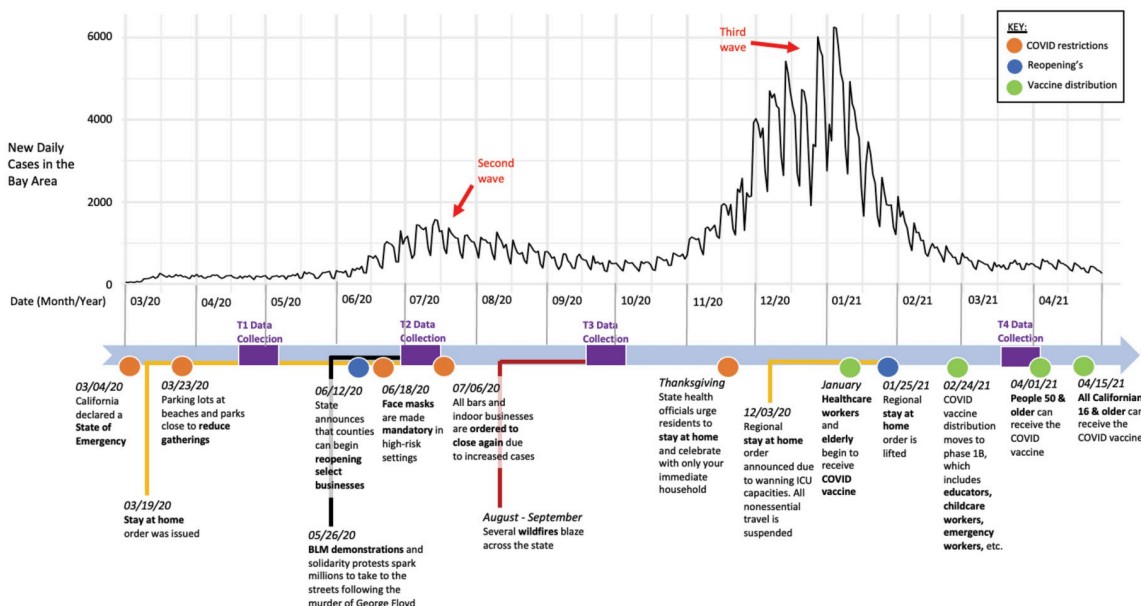

**Fig 1. Timeline of data collections, COVID-19 daily cases, and concurrent events.**

While the COVID-19 pandemic is a truly unprecedented global event that has impacted the lives of our participants, late spring and early summer 2020 also saw a wave of attention to the issues of racial injustice following the murder of George Floyd. As civil unrest marched on, the United States also endured an economic recession with record-high unemployment. A timeline of the study duration is presented in Fig 1, showing major events taking place throughout the study and the number of daily new cases of coronavirus in the San Francisco Bay Area.

## Participant sample

The Stanford WELL for Life data registry is composed of people at least 18 years of age and living in the US. Data registry participants were recruited through listservs, social media, and community partnerships [20]. We sent an email to all registry participants inviting them to join our longitudinal COVID-19 study. All individuals who agreed to participate, provided informed consent, and responded to the surveys at all 4 waves of data collection are included in this study (n = 701). Token incentives (e.g., reusable water bottles) were provided as a show of gratitude for participants' time and attention.

## Measures

**Benefit finding.** As a measure of benefit finding, we chose to use the Post-Traumatic Growth Inventory because it had been used effectively in previous studies of collective traumas [21, 22]. The Post-Traumatic Growth Inventory (PTGI) short-form is a 10-item questionnaire used to assess positive outcomes experienced by individuals who have been through a traumatic event [23]. Five types of possible benefits are measured by two items each: Relating to Others, New Possibilities, Personal Strength, Spiritual Change, and Appreciation of Life. Participants were asked to rate the extent to which they had experienced each as a result of the COVID-19 pandemic on a scale ranging from 1 = "I did not experience this" to 6 = "I experienced this to a very great extent."

To explore the prevalence of different types of benefit finding, we followed the suggestion of Tedeschi and colleagues [7]. We identified participants who answered either "I experienced this to a great extent" or "I experienced this to a very great extent" as having experienced significant benefit finding of the type described by each individual item. Additionally, we measured the overall prevalence of significant benefit finding as those who selected "…to a great extent" or "…to a very great extent" on at least one benefit finding questionnaire item.

Other studies have averaged all 10 inventory items to create a single overall continuous score [24, 25]. However, our factor analysis suggested using a 2-item scale for spiritual change and an 8-item scale for all other benefits (labeled general benefits). The Spiritual Benefits score and General Benefits score were calculated by taking the mean of the component items. Each score ranges from 1 to 6. The Cronbach alphas for the spiritual benefits scale and the general benefits scale at 1 month were 0.84 and 0.88, respectively. Similar Cronbach alphas were seen at all four timepoints. The distribution of the General Benefits variable approximated normal. However, the distribution of the Spiritual Benefits variable was highly skewed, with 60% or more of the study participants having responded "I did not experience this" at each of the four timepoints. Thus, for analysis purposes, we dichotomized the Spiritual Benefits variable (0 = those who responded "I did not experience this", 1 = all others).

**Well-being.** The WELL score is a multi-faceted comprehensive measure of well-being. Previous qualitative work identified domains of well-being and then survey questions were constructed and tested to measure these domains. Nineteen survey items were used in the WELL for Life registry to measure these domains of well-being: social connectedness, experience of positive emotions, experience of negative emotions, physical health, stress, resilience, purpose and meaning, sense of self, financial security, spirituality and religiosity, and exploration and creativity. A confirmatory factor analysis showed adequate measurement model fit. Fit was examined using the root mean square error of approximation (RMSEA = 0.05) and the comparative fit index (CFI = 0.96). The responses to each item were recoded to 0–100 points A mean score for all the items in a particular domain was calculated, and then a mean of the 11 domain scores was calculated to obtain a participant's WELL score. Therefore, scores could potentially range from 0 to 100 with higher scores indicating higher well-being.

The World Health Organization Well-Being Questionnaire (WHO-5) is a well-validated measure of positive well-being [26]. The measure consists of 5 items (e.g. "I have felt calm and relaxed") that ask participants to rate how often they have been feeling certain ways over the last two weeks on a scale from 0 = "At no time" to 5 = "All of the time." Scores were calculated through the summation of all five responses and multiplied by a factor of 4. Therefore, scores ranged from 0 to 100, where a higher score indicates more positive well-being.

**Other measures.** Social connectedness was measured with 2 questions adapted from the UCLA Loneliness Scale [27]. The questions asked participants, "During the last two weeks, how often did you feel that there were people you could talk to?" and "During the last two weeks, how often did you feel that there were people you could rely on?" Thus, this is a measure of the perception that one is part of a supportive network. To make the scoring consistent with the WELL Score, the responses were recoded to 0–100 points and, then a mean of the 2 questions was calculated. However, the distribution of the social connectedness variable did not approximate normal. Thus, we categorized participants into high, medium, and low social connectedness based on natural breaks in the distribution.

A measure of distress was adapted from the National Comprehensive Cancer Center Network Distress Thermometer, a well-known tool for identifying distress levels [28]. The item asks respondents to select "the number (0–10) that best describes how much distress [they] have been experiencing in the past week including today." A score of 10 indicates extreme distress and a score of 0 indicates no distress [29].

### Analysis plan

Linear and logistic regression modeling were used to examine the personal characteristics associated with benefit finding. Linear regression was conducted with the general benefits scale as the dependent variable and social connectedness, distress, and demographic variables as the predictors. Similarly, a logistic regression was conducted using the dichotomous variable for the experience of spiritual benefits (1 = any spiritual change, 0 = no spiritual change) as the dependent variable and using the same set of predictors. Demographic predictors included age, gender, educational attainment, and race.

To assess individuals' changes in benefit finding over time, the Reliable Change Index was utilized to calculate significant change [30–32]. Cut points derived from this procedure allowed for the categorization of participants into those who have decreased, increased, or remained unchanged in their general benefit finding between any 2 data collection timepoints. Reliable Change was calculated to be any change in benefit finding greater than or equal to 0.75.

To examine the relationship between well-being and general benefit finding over time, linear regression was conducted with subsequent well-being as the outcome. Previous well-being, previous general benefits score, and a set of dummy variables indicating whether the experience of general benefit finding had decreased, increased, or stayed the same since the previous time point were used as the predictors.

## Results

### Description of the sample

Table 1 shows the sociodemographic characteristics of the participants in terms of their age, gender, race, marital status, education level, employment status, and yearly income. The sample is predominantly female (78%), white (72%), and highly educated (54% with a graduate degree). Participants vary in age but tend toward the older age categories. Thus, 22% of the participants report being retired and only 26% have children living at home. It is also notable that less than 1% of the participants reported that anyone in their households had been diagnosed with COVID-19 during the year of the study.

### Type and level of benefit finding at different time points during the pandemic

Fig 2 presents the percentage of participants who responded that they experienced the type of benefit finding asked by an item to "a great extent" or "a very great extent" at each of the four time points. Having a greater appreciation for the value of one's life was the benefit most often experienced at all time points (17% to 20% of participants experienced this to a great or very great extent across all the time points). For nearly all the items, the highest percentages were at the final time point, 12 months following the shelter-in-place mandate. Benefit finding in the areas of experiencing new possibilities and perceiving personal strengths was experienced by a significantly higher percentage of participants at the 12-month mark than at the previous time points.

However, there are some notable exceptions to this pattern. For example, the highest percentage of participants reported that they "learned a great deal about how wonderful people are" at the beginning of the pandemic. This item exhibited a different pattern over time than did the other items. Spiritual change was experienced by only 5%–10% of participants at each of the data points, with no significant change over time.

The overall prevalence of significant benefit finding, measured as those who answered "I experienced this to a great extent" or "I experienced this to a very great extent" on at least one

**Table 1. Participants' sociodemographics.**

| | N (%) |
|---|---|
| Age (n = 701) | |
| • 18–29 | 76 (10.9%) |
| • 30–39 | 125 (17.8%) |
| • 40–49 | 106 (15.1%) |
| • 50–59 | 129 (18.4%) |
| • 60–69 | 140 (20.0%) |
| • 70+ | 125 (17.8%) |
| Gender (n = 698) | |
| • Female | 546 (78.2%) |
| • Male | 148 (21.2%) |
| • Non-binary | 4 (0.6%) |
| Race (n = 695) | |
| • White/Caucasian | 502 (72.2%) |
| • Asian/Pacific Islander | 161 (23.2%) |
| • Other | 32 (4.6%) |
| Marital Status (n = 701) | |
| • Married / Living with partner | 500 (71.3%) |
| • Previously married | 76 (10.9%) |
| • Single | 125 (17.8%) |
| Education Level (n = 696) | |
| • No college degree | 73 (10.5%) |
| • Bachelor's degree | 244 (35.1%) |
| • Post-Graduate | 379 (54.4%) |
| Employment Status (n = 674) | |
| • Working full time | 413 (61.3%) |
| • Working part time | 51 (7.6%) |
| • Temporarily laid off / unemployed | 13 (1.9%) |
| • Retired | 150 (22.2%) |
| • Homemaker / Student / Disabled / Other | 47 (7.0%) |
| Yearly Income (n = 670) | |
| • $0 - $49,999 | 59 (8.8%) |
| • $50,000 - $99,999 | 162 (24.2%) |
| • $100,000 - $149,999 | 155 (23.2%) |
| • $150,000 - $249,999 | 179 (26.7%) |
| • $250,000 - $499,999 | 88 (13.1%) |
| • $500,000 or more | 27 (4.0%) |
| Having Children at Home During COVID-19 (n = 701) | |
| • Those with children at home | 184 (26.2%) |
| • Those without children at home | 517 (73.8%) |

question, ranged from 34% to 42% for our participants throughout the pandemic. The lowest percentage occurred at the 6-month time point and the highest at the 12-month time point.

Table 2 presents the descriptive statistics for the major study variables at each of the study time points. The levels of spiritual benefits remained quite low across all time points. While the means for both of the benefit finding scales are quite similar across the first 3 data points,

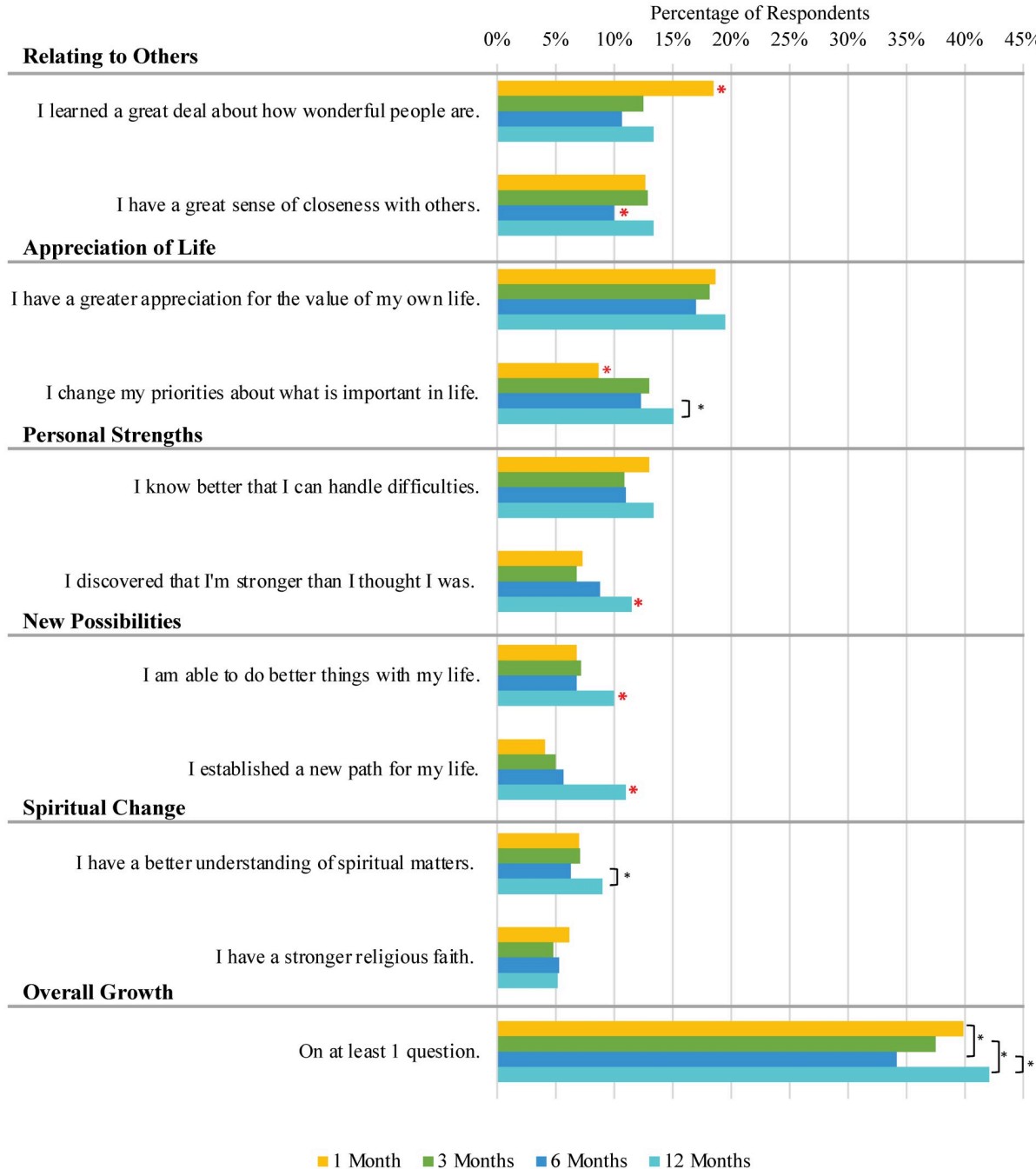

**Fig 2. Proportion of participants who answered "I experienced this to a great extent" or "I experienced this to a very great extent" on the benefit finding questions.** Note: The red asterisks represent a time point that is significantly different ($p < 0.05$) from all other time points, while the other asterisks show pairwise differences.

the means increase enough at the 12-month data point to elicit a significant F-test p-value generated by repeated measures ANOVAs. The well-being measures follow a similar pattern. Distress levels were notably higher at 6 months and lower at 12 months. Levels of social connectedness did not differ over time.

**Table 2. Descriptive statistics for major study variables at all four time points.** *Calculated by repeated measures ANOVA F-test.

| Study Variable | 1 Month | | 3 Months | | 6 Months | | 12 Months | | |
|---|---|---|---|---|---|---|---|---|---|
| | Mean | SD | Mean | SD | Mean | SD | Mean | SD | P-value* |
| Distress Thermometer | 3.81 | 2.49 | 3.81 | 2.44 | 4.26 | 2.64 | 3.43 | 2.62 | <0.01 |
| Well-being | | | | | | | | | |
| WELL Score | 70.40 | 14.51 | 69.97 | 13.99 | 68.98 | 14.70 | 71.35 | 14.52 | <0.01 |
| WHO-5 Score | 56.68 | 22.32 | 56.59 | 21.41 | 54.10 | 22.50 | 58.21 | 22.04 | <0.01 |
| Benefit Finding | | | | | | | | | |
| General Benefits | 2.49 | 1.05 | 2.54 | 1.04 | 2.51 | 1.06 | 2.68 | 1.09 | <0.01 |
| Spiritual Benefits | N | (%) | N | (%) | N | (%) | N | (%) | |
| Experienced to any extent | 267 | (38%) | 272 | (39%) | 257 | (37%) | 278 | (40%) | n.s. |
| Social Connectedness | N | (%) | N | (%) | N | (%) | N | (%) | |
| Low | 105 | (15%) | 121 | (17%) | 118 | (17%) | 106 | (15%) | n.s. |
| Medium | 296 | (43%) | 298 | (43%) | 285 | (41%) | 293 | (42%) | n.s. |
| High | 294 | (42%) | 278 | (40%) | 292 | (42%) | 294 | (43%) | n.s. |

Despite the small amount of aggregate change over time, individuals may still have been experiencing changes in benefit finding between time points. Fig 3 presents the percentage of participants who experienced increased, decreased or unchanged scores on the general benefits scale between the study time points. The majority of participants' experiences of benefit finding remained constant over time, but there was a sizable proportion of participants that experienced either an increase or decrease in general benefit finding between time points. Notably, fewer participants experienced a decrease in benefit finding between 6 and 12 months in comparison to the previous time period, and more participants experienced an increase in benefit finding between 6 and 12 months than during previous time periods.

## Cross-sectional associations between personal characteristics and benefit finding

Table 3 presents the results of a linear regression model predicting the experience of general benefits, using demographic variables, self-reported distress, and social connectedness as independent variables at the 1-month time point. Results from a binomial logistic regression predicting spiritual benefits (1 = experience of any, 0 = none) using the same independent variables are also presented. Participants who identified as Asian or Pacific Islander were more likely to experience benefit finding across both measures. Identifying as female was strongly associated with greater experience of general benefits, but the same relationship was not observed with spiritual benefits. Participants in the younger two age groups (18–29 and 30–39) were less likely to experience spiritual benefits, while participants in just the youngest age group (18–29) were less likely to experience general benefits. Educational attainment was negatively associated with both measures of benefit finding. Having children living at home was positively associated with general benefit finding but not with spiritual benefits. Having a supportive social network was more strongly positively associated with experiences of general benefits than with spiritual benefits. Lastly, greater experiences of distress during the last week were associated with greater benefit finding, with a stronger effect seen for general benefits. The 3 month, 6 month, and 12 month regression results presented similar findings with only minor differences, and can be found in the S1–S3 Tables.

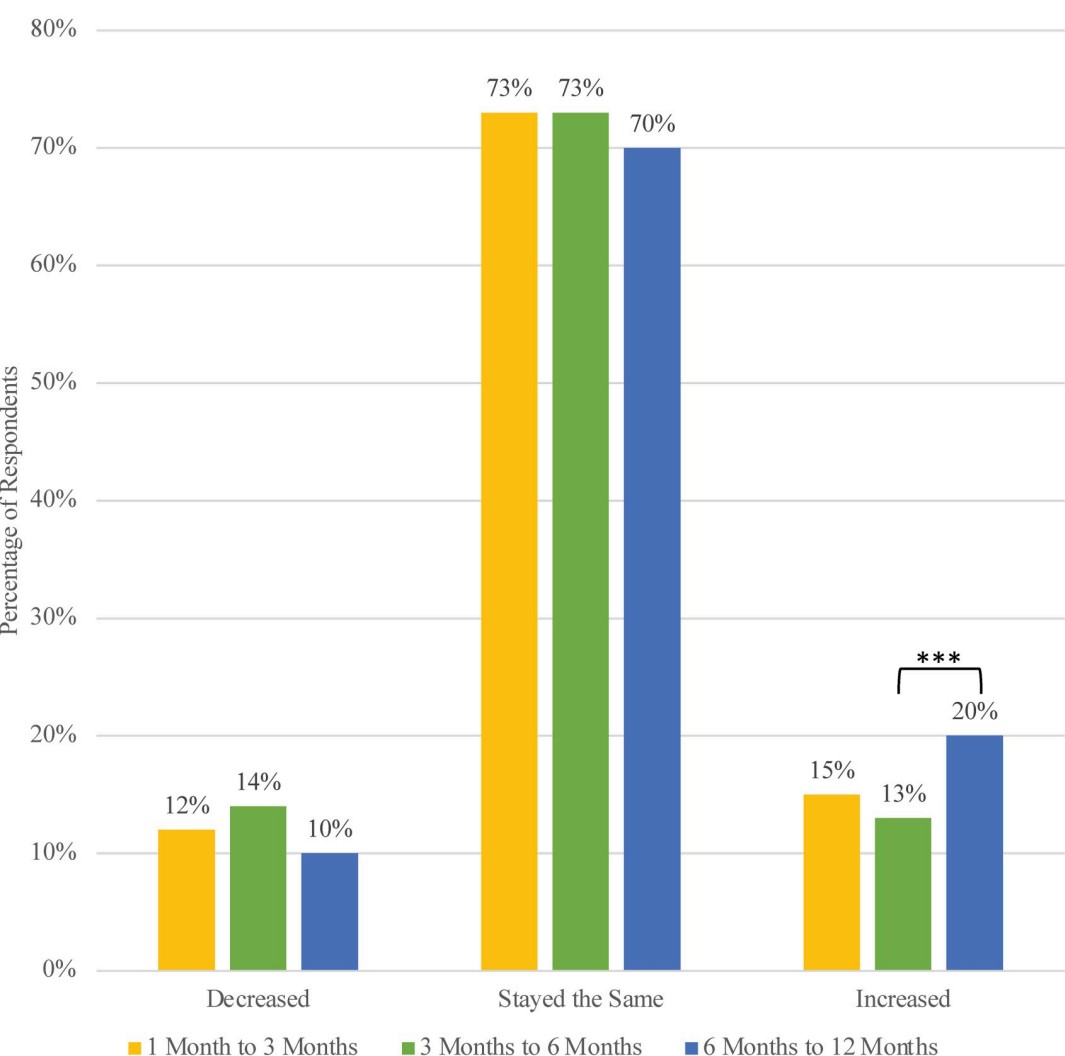

**Fig 3. Proportion of individuals' general benefits scores that increased, decreased, or remained unchanged between time points.** Note: ** p<0.01; *** p<0.001.

The extent to which the demographic sub-groups associated with higher levels of benefit finding were also experiencing higher levels of distress was also examined. At the 1-month time point, women reported significantly (p<0.01) higher levels of distress (M = 4.16, SD = 2.30) compared to men (M = 2.97, SD = 2.26). Additionally, adults who had children at home during COVID-19 reported significantly (p<0.01) higher levels of distress (M = 4.31, SD = 2.37) compared to those who did not have children at home (M = 3.75, SD = 2.33). For both educational attainment and race, distress did not significantly differ.

## Longitudinal associations between well-being and benefit finding

Cross-sectionally, general benefit finding and well-being are modestly but significantly positively correlated at each of the four time points (see S4 Table). Table 4 shows the results of longitudinal regression analyses using two different measures of well-being–the

**Table 3. Regression models predicting spiritual benefits and general benefits at 1 month.**

| | Spiritual Benefits Logistic Regression | | | General Benefits Linear Regression | | |
|---|---|---|---|---|---|---|
| | Odds Ratio | (95% CI) | P-value | Coefficient | SE | P-value |
| Intercept | 0.26 | (0.13–0.52) | <0.01 | 1.54 | 0.17 | <0.01 |
| Gender | | | | | | |
| Female | 1.14 | (0.76–1.73) | 0.53 | 0.39 | 0.10 | <0.01 |
| Male (ref.) | | | | | | |
| Race | | | | | | |
| Asian/Pacific Islander | 1.86 | (1.25–2.77) | <0.01 | 0.40 | 0.10 | <0.01 |
| Other | 1.26 | (0.58–2.66) | 0.54 | 0.25 | 0.18 | 0.17 |
| White/Caucasian (ref.) | | | | | | |
| Age (years) | | | | | | |
| 18–29 | 0.31 | (0.15–0.62) | <0.01 | -0.34 | 0.16 | 0.03 |
| 30–39 | 0.45 | (0.25–0.81) | <0.01 | -0.17 | 0.14 | 0.22 |
| 40–49 | 0.85 | (0.45–1.59) | 0.61 | -0.17 | 0.16 | 0.28 |
| 50–59 | 1.00 | (0.57–1.74) | 0.99 | -0.06 | 0.14 | 0.67 |
| 60–69 | 1.05 | (0.62–1.75) | 0.87 | -0.12 | 0.13 | 0.36 |
| 70+ (ref.) | | | | | | |
| Education | | | | | | |
| No college degree | 1.47 | (0.85–2.53) | 0.16 | 0.34 | 0.13 | 0.01 |
| Bachelor's degree | 1.49 | (1.04–2.12) | 0.03 | 0.19 | 0.09 | 0.03 |
| Graduate degree (ref.) | | | | | | |
| Having children at home during COVID-19 | | | | | | |
| With children | 1.33 | (0.88–2.03) | 0.18 | 0.21 | 0.10 | 0.04 |
| Without children (ref.) | | | | | | |
| Social connectedness | | | | | | |
| High | 1.53 | (0.93–2.57) | 0.10 | 0.44 | 0.12 | <0.01 |
| Medium | 1.33 | (0.82–2.20) | 0.26 | 0.25 | 0.12 | 0.04 |
| Low (ref.) | | | | | | |
| Distress thermometer | 1.08 | (1.01–1.16) | 0.03 | 0.06 | 0.02 | <0.01 |

Note: Spiritual Benefits is a dichotomous variable in which 0 represents those who responded "I did not experience this" and 1 represents all others.

Stanford WELL score and the WHO-5 well-being index. As described earlier, to predict well-being at a given subsequent time point, we used the well-being score from the previous time point, general benefits score from the previous time point, and whether there was an increase or decrease in benefit-finding from the previous time point as predictors. Results show that prior well-being scores were highly associated with subsequent well-being scores for both measures, indicating strong stability over time. The magnitude of the contribution of benefit finding variables to each of the regression models is small but statistically significant. Lagged effects of general benefits on subsequent well-being are not much in evidence. Those who decreased their benefit finding between 1 month and 3 months saw a significant decrease in well-being for both well-being measures. However, during the other study time periods, we see different results for the two well-being measures. For the WELL score, both the associations of decreases in benefit finding and increases in benefit finding with well-being are evident. For the WHO-5, increases in benefit finding are more strongly associated with well-being than are decreases in benefit finding.

**Table 4. Longitudinal regression models using general benefit finding to predict two measures of well-being.**

|  |  | 1 Month to 3 Months | | | 3 Months to 6 Months | | | 6 Months to 12 Months | | |
|---|---|---|---|---|---|---|---|---|---|---|
|  |  | Coefficient | SE | P-value | Coefficient | SE | P-value | Coefficient | SE | P-value |
| WELL Score | Intercept | 11.16 | 1.76 | <0.01 | 10.40 | 2.02 | <0.01 | 15.71 | 2.06 | <0.01 |
|  | WELL score | 0.83 | 0.02 | <0.01 | 0.87 | 0.02 | <0.01 | 0.78 | 0.02 | <0.01 |
|  | General Benefits | 0.18 | 0.29 | 0.53 | -0.06 | 0.33 | 0.85 | 0.82 | 0.35 | 0.02 |
|  | Increase in benefit finding* | 1.19 | 0.80 | 0.14 | 2.42 | 0.97 | 0.01 | 1.70 | 0.86 | 0.05 |
|  | Decrease in benefit finding* | -2.48 | 0.94 | <0.01 | -1.30 | 1.03 | 0.21 | -2.69 | 1.16 | 0.02 |
|  | $R^2$ | 0.74 |  |  | 0.70 |  |  | 0.66 |  |  |
|  | Delta $R^2$ ** | 0.004 |  | 0.02 | 0.004 |  | 0.02 | 0.007 |  | <0.01 |
| Well-Being Index (WHO-5) | Intercept | 20.50 | 2.66 | <0.01 | 13.58 | 2.64 | <0.01 | 22.86 | 2.70 | <0.01 |
|  | WHO-5 score | 0.66 | 0.03 | <0.01 | 0.75 | 0.03 | <0.01 | 0.68 | 0.03 | <0.01 |
|  | General Benefits | 0.03 | 0.60 | 0.96 | 1.16 | 0.60 | 0.05 | 0.32 | 0.63 | 0.61 |
|  | Increase in benefit finding* | 1.98 | 1.71 | 0.25 | 5.00 | 1.77 | <0.01 | 5.82 | 1.57 | <0.01 |
|  | Decrease in benefit finding* | -4.05 | 1.94 | 0.04 | -2.66 | 1.81 | 0.14 | -2.11 | 2.11 | 0.32 |
|  | $R^2$ | 0.49 |  |  | 0.55 |  |  | 0.49 |  |  |
|  | Delta $R^2$ ** | 0.005 |  | 0.07 | 0.008 |  | <0.01 | 0.012 |  | <0.01 |

Note: All models were controlled for gender, age, race, education, and whether they had children at home during the COVID-19 pandemic.

*Both the increase and decrease in benefit finding variables coded (0,1) were compared to a reference group of individuals whose benefit finding scores stayed the same between the respective time points.

**The comparison model excluded the following independent variables: General Benefits, Increase in benefit finding, Decrease in benefit finding.

## Discussion

The aim of this study was to understand the experience of benefit finding during the course of the COVID-19 pandemic. Prior research on benefit finding has largely been conducted after the traumatic event has ended. As we emerge from the turbulence of the COVID-19 pandemic, understanding benefit finding in the midst of a collective trauma is both essential and novel.

At no point during our study did more than 20% of the participants strongly experience any one specific type of benefit finding, with some types of benefit finding experienced by fewer than 10%. However, between 34% and 42% experienced some type of benefit finding at various points during the pandemic. It is difficult to compare these percentages with other studies because various methods have been used to designate whether a person has experienced benefit finding [7]. In the extant literature, the percentage of participants in any given study that are experiencing benefit finding has ranged from 3% to 98% [9]. Additionally, mean scores across studies are difficult to compare since the measures used often included different survey items using different response scales. Therefore, it is quite ambitious to draw any conclusion as to how the levels of benefit finding in the current study compare with those in previous studies.

However, quite consistently across studies, the standard deviations of benefit finding scores are quite large. In studies examining benefit finding during the COVID-19 pandemic in Greece, China, and the US, standard deviations were approximately 54%, 39%, and 78% of the relevant means [33–35]. In studies exploring traumas such as earthquakes, the experiences of Syrian refugees, or the loss of a child, standard deviations also were consistently high at 34%, 59%, and 41% of the mean scores, respectively [21, 22, 36]. Another way to conceptualize the extent of variation is to compare the standard deviation to the potential range of the score. The

standard deviations were typically between 20% to 30% of the possible range. Thus, there is much variation in the extent to which individuals experience benefit finding.

Time elapsed since the trauma has been shown to be a positive predictor for benefit finding [37]. As individuals have more time to reflect and recover, they may be more able to extract wisdom, strength, and overall positive changes from the process of experiencing the trauma. However, in our study, the different data collection time points do not reflect time since a time-bounded trauma has occurred, but rather the time since the beginning of an ongoing collective trauma, as well as the changing nature of the pandemic.

The 12 months of this study's duration were marked by cascading collective traumas, associated with a rise in mental health problems across the United States, leading to pleas for policymakers to support community mental health in an effort to "strengthen the social fabric and ease the mental and physical health burden of these trying times" [38]. In previous studies, the number of traumas experienced or the amount of trauma-related exposure has been associated with higher levels of benefit finding [14, 39]. In this study, there are so many changes and potential traumas occurring that it is difficult to make clear interpretations about the cascading traumas and associations with other study variables. For example, at the six month data collection during which participants were experiencing the beginning of a spike in COVID-19 cases, the effects of wildfires, and continuing political unrest due to racial inequities, we see the highest levels of distress (see Table 2) and the highest proportions of those who decreased and the lowest proportions of those who increased in terms of general benefit finding (refer to Fig 3). However, these differences between timepoints are small in magnitude. Once vaccine availability entered the picture, distress significantly decreased and benefit finding and well-being exhibited small but significant increases. Very little is known about the long-term effects of cascading collective traumas like those experienced during this study, and our work is only the beginning of what we must understand in order to inform action.

Additionally, our findings show positive cross-sectional associations between experiences of distress and benefit finding, which contradicts some previous findings showing that distress had little or no association with benefit finding [8, 40]. In a study of post-traumatic growth of breast cancer patients in treatment and early survivorship, while cancer-specific stress and general anxiety were related to higher post-traumatic growth, overall general distress had minimal association with benefit finding. A complex rendering of the relationship between distress and benefit finding emerged from the 2006 meta-analysis [8]. Benefit finding was unrelated to global distress. Time since trauma was a significant moderator of the association between benefit finding and several mental and physical health outcomes. Most notably, benefit finding was related to more distress only when less than 2 years had passed since the traumatic event took place. Since our data collection not only occurred within two years of the traumatic event, but actually during the turbulence of the COVID-19 pandemic, it is reasonable to conclude that distress acts as a catalyst for benefit finding when the trauma is fresh in the minds of individuals.

Identifying as female, having Asian origins, living with children, lower educational attainment, and having supportive social networks were positively significantly associated with benefit finding at all four timepoints in our data. Some of these associations may be due to a concentration of exposure to trauma-related events in specific sub-groups of the population. For example, in the United States, having Asian origins has been associated with elevated levels of racial discrimination and violence during the pandemic. A survey conducted in April 2021 of a representative US sample reported that 81% of Asian American adults reported that violence against Asian Americans has been rising [41]. Obviously, this could be experienced as an additional trauma. The gender effect might be similarly explained. Many previous studies of collective traumas such as earthquakes, forced displacement from

war, and COVID-19 in other countries, have shown gender differences in benefit finding with women exhibiting significantly higher scores [12, 22, 33]. Women, commonly the primary caregiver in US families, were more likely to have gone without medical care since SIP, to have lost their jobs due to the pandemic, or to have taken unpaid sick leaves to care for their children [42, 43]. Thus, the severity of the trauma of the pandemic may be greater for women than for men.

The same logic might apply to those with children at home and to those with lower educational attainment. Parents with children at home were more likely to have increased familial responsibilities due to widespread closures of schools and other childcare facilities. Lastly, individuals with lower educational attainment have fared substantially worse throughout the pandemic. In May 2020, the unemployment rate for individuals in the US with a high school diploma or less rose 12.0 percentage points compared to the 5.5 percentage points for individuals with a bachelor's degree [44]. It is possible that these sub-groups in our study who experienced higher levels of benefit finding were also the individuals who were facing heightened exposure to personal aspects of the collective trauma and/or additional traumas. However, further research is needed to test this hypothesis.

In addition to more severe exposure, higher levels of distress may play a role in increasing benefit finding for these sub-groups. As reported earlier, women and participants with children living at home reported high levels of distress. However, as shown in Table 3, the relationship between these sub-groups and benefit finding is still present when the contribution of distress is controlled for.

Our findings suggest a positive association between a supportive social network and the experience of benefit finding. Research has consistently found a positive relationship between social support and physical and mental health [45]. Particularly, strong social connections help lessen stress reactions in various situations. In the context of trauma and adverse life events, supportive social relationships have been shown to help individuals suffering from childhood abuse [46], mothers of children with chronic physical conditions [47], and cancer patients [13]. In the collective trauma literature, social support is consistently significantly associated with increased benefit finding. In a study of Gulf War Veterans, post-deployment social support from family, friends, coworkers, employers, and the community was the only significant predictor of benefit finding in their final model [48]. Jia and colleagues found that social support was predictive of subsequent post-traumatic growth in a population of Chinese adults affected by the Wenchuan Earthquake [21], supporting a causal effect of social support on benefit finding. While the specific types of social support needed to catalyze the experience of benefit finding are not yet known, there is sufficient evidence to suggest that focusing on enhancing one's supportive network may increase benefit finding.

Lastly, research on the association between benefit finding and well-being has had inconsistent results [8, 49]. In this study, the cross-sectional associations were consistently positive but small in magnitude across the four data points. Our longitudinal models also showed associations of small magnitude—as benefit finding increased or decreased over time, so did well-being as measured by both the WELL score and the WHO-5. The causal direction of the associations cannot be assessed from our analysis. Increases in benefit finding were more strongly associated with changes in the WHO-5 than with the WELL score. This may be due to the WHO-5 emphasis on positive emotions as opposed to the broad content of the WELL score. Further research is needed to more fully understand the role of benefit-finding in the process of responding to traumas. For example, post-traumatic growth, or benefit finding, has shown to have a moderating effect on the relationship between post-traumatic stress and both depression and quality of life among breast cancer survivors [49]. Similar studies of responses to collective traumas would be informative.

## Limitations

The most significant limitation of our study is that it is a non-representative sample that includes high proportions of individuals who are female, highly educated, wealthy, racially identify as white or Asian/Pacific Islander, and reside in the San Francisco Bay Area. Thus, further research is necessary to assess these same research questions among individuals of other sociodemographic backgrounds. Additionally, the PTGI only measures a specific set of benefits that may exclude additional benefits experienced during the pandemic. Specifically, our measure of benefit finding failed to assess benefits that have arisen due to the specific nature of the COVID-19 pandemic such as additional time with family, the elimination of lengthy commutes for many, and an enhanced appreciation for nature and outdoor activities [50, 51]. Lastly, indicators of severity of exposure to the collective trauma were not included in the study and would have offered an enhanced understanding of the determinants of benefit finding and well-being.

## Implications for promoting well-being

Even though the association between benefit finding and well-being does not seem to be strong, increasing people's opportunities and abilities for experiencing benefits is likely to play a role in reducing stress and enhancing well-being. Some previous research has demonstrated that the experience of benefit finding can be increased, for example in response to work-related traumas [52] and in response to cancer [53]. However, further research is needed to assess the effectiveness of strategies to enhance benefit finding in response to collective traumas. This study suggests that a supportive social network and time to reflect may increase the likelihood of benefit finding and contribute to its positive effects.

## Conclusion

Benefit finding has been widely studied weeks, months, and years after a personal traumatic event, but little research has been conducted on benefit finding while a collective trauma is still occurring. As COVID-19 persists and additional variants continue to spread around the world, there is a heightened need to understand concurrent benefit finding and its potential to promote well-being.

## Supporting information

**S1 Table. Regression models for predicting spiritual benefits and general benefits at 3 months.**
(TIF)

**S2 Table. Regression models for predicting spiritual benefits and general benefits at 6 months.**
(TIF)

**S3 Table. Regression models for predicting spiritual benefits and general benefits at 12 months.**
(TIF)

**S4 Table. Cross-sectional correlations between the general benefits score and two measures of well-being.** *The General Benefits Score at 1 month, 3 months, 6 months, and 12 months was used, respectively.
(TIF)

## Acknowledgments

We thank Katy Peng for assistance in compiling the dataset. We also thank Ann Hsing for her tremendous support and guidance.

## Author Contributions

**Conceptualization:** Catherine A. Heaney.

**Formal analysis:** Jessie B. Moore, Katharine C. R. Rubin, Catherine A. Heaney.

**Investigation:** Catherine A. Heaney.

**Methodology:** Jessie B. Moore.

**Project administration:** Catherine A. Heaney.

**Software:** Jessie B. Moore.

**Supervision:** Catherine A. Heaney.

**Validation:** Jessie B. Moore, Catherine A. Heaney.

**Visualization:** Jessie B. Moore, Katharine C. R. Rubin.

**Writing – original draft:** Jessie B. Moore, Katharine C. R. Rubin, Catherine A. Heaney.

**Writing – review & editing:** Jessie B. Moore, Catherine A. Heaney.

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
