## [Decision Letter · Decision Letter 0]

15 Feb 2023

PONE-D-22-33849Benefit finding and well-being over the course of the COVID-19 pandemicPLOS ONE

Dear Dr. Moore,

Thank you for submitting your manuscript to PLOS ONE. After careful consideration, we feel that it has merit but does not fully meet PLOS ONE’s publication criteria as it currently stands. There are issues with the conceptualization and statistical handling of the data. Therefore, we invite you to submit a revised version of the manuscript that addresses the points raised during the review process.

We look forward to receiving your revised manuscript.

Kind regards,

Qin Xiang Ng, MBBS, MPH

Academic Editor

PLOS ONE

Journal Requirements:

2. In the Methods section of your manuscript, please provide additional details regarding participant consent. In the ethics statement in the Methods and online submission information, please ensure that you have specified what type you obtained (for instance, written or verbal, and if verbal, how it was documented and witnessed). If your study included minors, state whether you obtained consent from parents or guardians. If the need for consent was waived by the ethics committee, please include this information.

3. Please ensure that you include a title page within your main document. We do appreciate that you have a title page document uploaded as a separate file, however, as per our author guidelines (http://journals.plos.org/plosone/s/submission-guidelines#loc-title-page) we do require this to be part of the manuscript file itself and not uploaded separately.

Could you therefore please include the title page into the beginning of your manuscript file itself, listing all authors and affiliation.

"Initial foundational funding for the Stanford Wellness Living laboratory (WELL) was provided by Amway via an unrestricted gift through the Nutrilite Health Institute Wellness Fund to Stanford University. Preparation of this manuscript was supported in part by the Stanford Thailand Research Consortium."

Reviewers' comments:

Reviewer's Responses to Questions

**Comments to the Author**

1. Is the manuscript technically sound, and do the data support the conclusions?

Reviewer #1: Yes

Reviewer #2: Partly

2. Has the statistical analysis been performed appropriately and rigorously? 

Reviewer #1: Yes

Reviewer #2: No

3. Have the authors made all data underlying the findings in their manuscript fully available?

Reviewer #1: Yes

Reviewer #2: Yes

4. Is the manuscript presented in an intelligible fashion and written in standard English?

Reviewer #1: Yes

Reviewer #2: Yes

5. Review Comments to the Author

Reviewer #1: This manuscript examines if benefit finding is associated with mental well-being over the COVID-19 pandemic in a large sample of American participants in California. Overall, the manuscript is well written and its findings should be of interest to the general readership of PLoS ONE. I believe that it should be appropriate for publication after addressing several concerns that I have.

1. I think the manner in which the participants are classified as having increments, decrements, or no change between described on p. 10 needs more justification. Why is half a point chosen as a cut-off? Any prior evidence to suggest that this cut-off value is appropriate? Have the authors considered the Reliable Change Index (Jacobson and Truax, 1991)? This index allows for the estimation of reliable change (after accounting for measure unreliability) over time for an individual. The authors would be able to derive the percentage of people who have decreased, increased, or remained unchanged in benefit finding.

2. Table 3, for the results concerning the logistic regression (spiritual benefits), the authors should also present the odds ratios for the individual predictors. Odds ratios are more intuitive to interpret in this context than are regression coefficients.

3. While the results presented in Table 4 are interesting, the use of the categorical variable of benefit finding is a limitation. Why not simply use the original measure (as a continuous variable) so that statistical information is not lost when one transform a continuous variable to a categorical one? In addition, the analyses presented in Table 4 resemble a cross-lagged panel model (CLPM). I would encourage the authors to conduct the CLPM as an integrative approach rather than conducting multiple regression analyses. For example, the longitudinal associations between benefit finding and WELL scores can be modelled over the 4 time points. A separate model can be conducted with benefit finding and WHO-5 scores. The cross-lagged associations will address the question of whether prior benefit finding would predict subsequent well-being, or vice versa.

4. In the Discussion, the authors suggest that being women, Asian, having lower educational attainment status, and less social support are being more exposed to the negative and traumatic effects of COVID-19, and hence the observed heightened benefit finding. The authors offer this possibility (i.e., being more exposed to trauma) and thinks that future is needed. Hence, I find it bizarre that in this study, the authors have not tracked people’s exposure to trauma, or some proxy of trauma. This would be a major study limitation. The only measure that is close to trauma exposure would be the one-item measure of “distress”. Perhaps this distress can be correlated with all these predictor variables, and see if the proposed mechanism is plausible (e.g., having lower educational attainment status is associated with more distress)? Rather than speculations in the Discussion, there is some avenue for the authors to test some of these assertions. Granted, a distress measure is not the same as trauma exposure, but at least it could provide the authors with some support of their assertions if support is found.

In conclusion, I think there is promise in this manuscript. The findings are interesting, and have potential practical implications. Hence, I encourage the authors to consider the abovementioned issues in their revision.

Reviewer #2: 1. Overview

This study aimed to estimate the prevalence of benefit finding behavior in the course of COVID-19 pandemic, describe its pattern of variation, and examine its effect on individual well-being. It tested four hypotheses below using data from a questionnaire survey of a sample of Californians:

H1: individuals that report stronger support networks will be more likely to experience subsequent benefit finding.

H2. people who report higher levels of distress (regardless of whether the source of the distress is personal or societal) will be more likely to experience subsequent benefit finding

H3. amount of benefit finding will increase as the pandemic progresses and individuals have more opportunity to gain insight about their experiences.

H4: a positive relationship between benefit finding and subsequent well-being, particularly for the latter measure of well-being.

2. Specific comments

This study can potentially contribute to the body of knowledge on the factors associated with benefits finding and the effect of benefit finding on well-being. However, there are major conceptual and methodological issues (see below) that the authors need to address to improve the credibility of the study findings.

2.1. Introduction

The author claimed that the COVID-19 pandemic to be a collective trauma (Line 21) citing Tedeschi et al.’s definition of trauma “a highly stressful and challenging life altering event” (Lines 22) as reference for the concept of collective trauma. However, the cited definition is unable to distinguish collective trauma from personal trauma such as a fatal disease. The authors are advised to check out Kai Erikson’s definition of collective trauma which draws an intentional and clear distinction from personal drama: “a blow to the basic tissues of social life that damages the bonds attaching people together and impairs the prevailing sense of communality” (Erikson, 1976, p153). It is not apparent that COVID-19 meets the definition of a collective trauma by this definition, although it is apparently a collective stressor. The authors need to reconsider the validity of conceptualising COVID-19 as a collective trauma and provide strong and convincing justification for their conceptualisation.

2.2. Methods:

• Sampling method: The authors indicated that the study participants were recruited from an online registry—Stanford WELL for Life initiative (Lines 111-112). This is the sampling framework. It is important to describe the sampling procedure too i.e., how participants were selected from this registry, for example, by random sampling method?

• Selection of measurement scale for benefit finding: The authors ought to provide theoretical or methodological justification for their choice of PTGI as the measurement of the key concept of benefit finding over competing measurement tools such as the Benefit Finding Scale (Tomich & Helgeson, 2004) and the Perceived Benefit Scale (McMillen & Fisher, 1996).

• Dichotomisation of PTGI scale: The authors dichotomized participants into those who answered either “I experienced this to a great extent” or “I experienced this to a very great extent” as having experienced significant benefit finding and all others as not having experienced benefit finding for each individual item and citied Tedeschi et al.’s work as precedence. They ought to discuss critically the pros and cons of this practice and provide strong justification for its adoption.

• Inconsistency of measurement level for the same construct: In Lines 206-208, the authors stated that a logistic regression was conducted using a dichotomous variable for the experience of spiritual benefits (1=any spiritual change, 0=no spiritual change) as the dependent variable. However, in the earlier part of the manuscript (Lines 166-168), the authors had stated that the Spiritual Benefits score was calculated by taking the mean of the component items. Each score ranges from 1 to 6. It is unclear how a mean score of two 6-point Likert scales became a dichotomous measure. The authors ought to reconcile this inconsistency.

• Categorisation of continuous variables: The authors stated that the changes of PTGI scores over time were categorized into three groups signifying whether the experience of benefit finding had decreased, increased, or stayed the same from the previous time point. Scores were considered to have decreased if they had fallen by more than half a point, and to have increased if they had risen by more than half a point. If scores changed by less than half a point, they were considered to have stayed the same (Lines 212-216). Categorising continuous measures is a problematic practice. The authors are advised to refer to the following articles on the cost of dichotomisation and reconsider their analytical decision or at least discuss the cost of this decision.

o Altman DG, Royston P. The cost of dichotomising continuous variables. BMJ. 2006 May 6;332(7549):1080. doi: 10.1136/bmj.332.7549.1080. PMID: 16675816; PMCID: PMC1458573.

o Royston P, Altman DG, Sauerbrei W. Dichotomizing continuous predictors in multiple regression: a bad idea. Stat Med. 2006 Jan 15;25(1):127-41. doi: 10.1002/sim.2331. PMID: 16217841.

2.3. Findings

The authors found that for the WELL score, decreases in benefit finding are associated with

decreases in well-being; for the WHO-5, whereas increases in benefit finding are associated with

increases in well-being (Lines 311-313). As WELL and WHO-5 are measures of the same construct well-being, these seemingly paradoxical findings need to be explained in the discussion section.

2.4. Minor Issues:

• Lines 173-174, the meaning of “a strong confirmatory factor analysis (CFA)” is unclear.

• Line 174: “the score is composed of 19 items”, did the authors mean “scale” by the word “score”?

6. PLOS authors have the option to publish the peer review history of their article (what does this mean?). If published, this will include your full peer review and any attached files.

Reviewer #1: No

Reviewer #2: No

---

## [Author Response · Author response to Decision Letter 0]

15 Apr 2023

See attached document (response to reviewers).

---

## [Decision Letter · Decision Letter 1]

26 Jun 2023

Benefit finding and well-being over the course of the COVID-19 pandemic

PONE-D-22-33849R1

Dear Dr. Moore,

We’re pleased to inform you that your manuscript has been judged scientifically suitable for publication and will be formally accepted for publication once it meets all outstanding technical requirements.

Kind regards,

Qin Xiang Ng, MBBS, MPH

Academic Editor

PLOS ONE

Additional Editor Comments (optional):

Reviewers' comments:

Reviewer's Responses to Questions

**Comments to the Author**

1. If the authors have adequately addressed your comments raised in a previous round of review and you feel that this manuscript is now acceptable for publication, you may indicate that here to bypass the “Comments to the Author” section, enter your conflict of interest statement in the “Confidential to Editor” section, and submit your "Accept" recommendation.

Reviewer #1: All comments have been addressed

2. Is the manuscript technically sound, and do the data support the conclusions?

Reviewer #1: Yes

3. Has the statistical analysis been performed appropriately and rigorously? 

Reviewer #1: Yes

4. Have the authors made all data underlying the findings in their manuscript fully available?

Reviewer #1: Yes

5. Is the manuscript presented in an intelligible fashion and written in standard English?

Reviewer #1: Yes

6. Review Comments to the Author

Reviewer #1: The authors have taken into account most of my previous comments and addressed them adequately. I think that the revised manuscript has improved in the statistical treatment of the data, and the interpretation of the results. This is a paper worthy of publication. Congrats to the authors!

7. PLOS authors have the option to publish the peer review history of their article (what does this mean?). If published, this will include your full peer review and any attached files.

Reviewer #1: No

---

## [Editor Report · Acceptance letter]

18 Jul 2023

PONE-D-22-33849R1 

Benefit finding and well-being over the course of the COVID-19 pandemic 

Dear Dr. Moore:

I'm pleased to inform you that your manuscript has been deemed suitable for publication in PLOS ONE. Congratulations! Your manuscript is now with our production department. 

Kind regards, 

on behalf of

Dr. Qin Xiang Ng 

Academic Editor

PLOS ONE